# Attitudes towards Violence in Adolescents and Youth Intimate Partner Relationships: Validation of the Spanish Version of the EAV

**DOI:** 10.3390/ijerph18020566

**Published:** 2021-01-12

**Authors:** Javier Ortuño-Sierra, Andrea Gutiérrez García, Edurne Chocarro de Luis, Julia Pérez-Sáenz, Rebeca Aritio-Solana

**Affiliations:** Educational Sciences Department, University of La Rioja, 26004 Logroño, Spain; andrea.gutierrezg@unirioja.es (A.G.G.); eljavibueno@hotmail.com (E.C.d.L.); julia.perez@unirioja.es (J.P.-S.); rebeca.aritio@unirioja.es (R.A.-S.)

**Keywords:** EAV, adolescents, psychometric properties, intimate partner violence, attitudes

## Abstract

The main purpose of the present study was to analyze the psychometric properties of the Attitudes Scale Towards Violence (*Escala de Actitudes hacia la Violencia, EAV*) in adolescents. The EAV is a questionnaire devoted to assess attitudes towards violence. Additionally, the relationship between EAV and violence manifestations and depressive symptoms was analyzed. The final sample comprised a total of 1248 students in a cross-sectional survey. The EAV, the Modified Conflict Tactics Scale (M-CTS), and the Reynolds Adolescent Depression Scale (RADS) were used. The analysis of the internal structure of the EAV yielded a two-factor structure as the most adequate. The EAV scores showed measurement invariance across gender and age. The McDonald’s Omega was 0.862 and 0.872 for the two hypothesized factors. Furthermore, self-reported attitudes towards violence were associated with violence manifestations both as a victim and as a perpetrator and depressive symptoms. These results support that the EAV is a brief and easy tool to assess self-reported violence attitudes in intimate partner relationships in adolescents from the general population. The assessment of these attitudes, and its associations with violence and depressive manifestations, may help us to enhance the possibility of an early identification of adolescents potentially at risk for suffering violence as a victim or as a perpetrator.

## 1. Introduction

An increasing interest is being devoted to intimate partner violence (IPV), due to the severe physical, psychological, and social consequences associated, as well as the growing prevalence through the world [1,2,3,4]. This specific kind of violence that includes physical, verbal, and sexual violence has become a serious and prevalent problem, not only in adulthood but also in adolescence and into emerging adulthood [5,6,7]. For instance, The WHO Multi-country Study on Women’s Health and Domestic Violence against women revealed that women between 15 and 49 years old had suffer some kind of partner violence in percentages that ranged from 13 to 61 percent [8]. These rates are worrying, especially taking into account that some of the mental and behavioural problems, as well as violence manifestations that develops during adolescence, tend to perpetuate to adulthood [9,10,11]. This type of violence, involving an actual or former partner, is at this time the more prevalent way of violence affecting women, being a major public health problem. This situation is affecting women, and has become a major cause of social disengagement, economy burden, and mental health problems, including depression, substance abuse, or trauma [12,13,14,15]. Thus, nowadays, it is possible to talk of a global epidemic of violence against women, with a total life prevalence in the world of one third of women that have experience some type of violence related to a romantic relationship [16,17,18].

Worth noting, although most of the victims addressing IPV are females [6,17,19], it is known that males are also victims of females perpetrators, with a total of up to in between 20 to 30 percent of men recognizing being victims of this kind of behaviours in heterosexual relationships [20]. Moreover, this problem is also affecting to same-sex relationships [20,21]. In the case of adolescents, recent data suggest that IPV is affecting more than half of those that have engaged or that have experienced a relationship either as a victim or as perpetrator [2,22]. This is even more alarming considering that teenagers with maltreatment history have a greater risk of being involve in a relationship with IPV both as a victim and as a perpetrator [22,23].

Taking into account the before mentioned, it seems relevant to devote resources aimed to understand the inner mechanisms that is behind IPV. Thus, understanding the attitudes towards IPV that adolescents show could be relevant. Different previous studies have focused on the violence behaviours, but not on the attitudes, nor in the relationship between these attitudes and IPV. Among the different instruments devoted to assess these problematics, we can find the Attitude Scale towards Intimate Violence (*Escala de Actitudes hacia la Violencia Intima*, EAV) [24]. The EAV encompass ten items addressing the degree to which the individual considers appropriate the use of violence towards the partner in different situations. There is, thus, a general question asking: *In what circumstances do you consider the use of intimate partner violence justifie?* And then, ten different options such as: *when a member of the couple is unfaithful* (ítem 1) or *when a member of the couple disqualifies the other in front of his/her family* (ítem 3). 

Nonetheless, the psychometric properties of this recent version have not been, yet, reported. Therefore, a question needs still to be solve. Is it possible to use the EAV in its Spanish version as an instrument with adequate evidences of reliability of the scores and validity? Moreover, recent research reveals that different variables, including education, attitudes, and ideas about gender roles and expectations of a relationship are connected to IPV [17]. With this regard, and considering that the ideas and attitudes of adolescents towards IPV could be related to emotional symptoms [13,14], new attention is being devoted to the quality of romantic relationships and to this, in order to promote healthy relationships [25].

Considering the previous background, and the fact that the EAV whose psychometric properties has not been yet validated in Spanish population, outside South America, the main goal of the present study was, therefore, to analyze the psychometric properties of the EAV in a large sample of Spanish adolescents. We, thus, gather evidences about the structure of the questionnaire, study the measurement invariance (MI) by gender and age, and analyze evidences of validity with external variables. We hypothesized that a one-dimensional model would reveal adequate goodness-of-fit indices and that this factorial structure would be invariant across gender and age. In addition, we hypothesized that measures of EAV would be related to violence in the context of intimate relationships and depressive symptoms.

## 2. Method

### 2.1. Participants

In order to obtain a representative community sample, we recruited participants from different cities and different types of secondary schools (e.g., public, funded, and private) and vocational/technical schools belonging to Navarra and La Rioja (two regions located at the north of Spain). Both rural and urban areas were represented, as well as a range of socioeconomic levels. We recruited students from ten schools, including educational and training centres. The initial sample included 1305 students, and we discarded data from participants who presented: (a) omissions of any demographics or items without responding (*n* = 37); and (b) scores in the range of outliers (*n* = 20) (e.g., scores higher than 2.5 standard deviations in the subscales of the measures used). The final sample consisted of 1248 students, of which 483 were male (38.7%). The age of the participants ranged from 13 to 21 years-old (*M* = 16.12 years; *SD* = 2.12). The age distribution of the sample was the following: 13 years (*n* = 65; 5.2%), 14 years (*n* = 216; 17.3%), 15 years (*n* = 336; 26.9%), 16 years (*n* = 231; 18.5%), 17 years (*n* = 147; 11.8%), 18 years (*n* = 129; 10.3%), and 19–21 years (*n* = 92; 7.3%).

### 2.2. Instruments

#### 2.2.1. The Attitudes Scale towards Intimate Violence (Escala de Actitudes Hacia la Violencia Íntima (EAV) 

The EAV [24] is an instrument devoted to assess attitudes towards violence in the context of intimate and romantic relationships. The EAV is composed of 10 items in a Likert response format with five options (1 = “totally disagree”, 2 = “disagree”, 3 = “neutral”, 4 = “agree”, and 5 = “totally agree”. The items ask about under which circumstances the use of violence is justified in a relationship. For example, the use of violence is justified “When a member of the couple insult to the other” or “When one member of the people does not agree to have sexual intercourse”. The scale has shown evidences of internal consistency of the scores in previous studies with alpha values over 0.90 [24].

#### 2.2.2. The Modified Conflict Tactics Scale (M-CTS) 

The M-CTS [26] is one of the most widely used instrument to measure the way in which individuals deal issues with their partners. It addresses behaviours when arguing with the actual partner or the most recent relationship. The M-CTS is composed of 18 bidirectional items addressing behaviours as victim and aggressor. The items are in a 5-point Likert scale ranging from 1 (*never*) to 5 (*very often*). The validated Spanish version was used in the present study [27]. The M-CTS revealed adequate evidences of internal consistency of the scores in the present study with a McDonald’s Omega value of 0.82. In addition, CR value was 0.823 and AVE was 0.713. 

#### 2.2.3. The Reynolds Adolescent Depression Scale (RADS) 

The RADS [28] assesses the severity of depressive symptoms in adolescents. It is composed of 30 items in a Likert response format with four options (1 = “almost never”, 2 = “hardly ever”, 3 = “sometimes”, 4 = “most of the time”). The RADS encompasses four empirically derived scales: Anhedonia, Somatic complaints, Negative self-evaluation, and Dysphoric mood. The validated Spanish version of the RADS was used in the present study [29]. The RADS revealed adequate evidences of internal consistency of the scores in the present study with a McDonald’s Omega value of 0.84. The CR (0.815) and AVE (0.727) values were also adequate.

### 2.3. Procedures

The questionnaires were administered collectively, in groups of 10 to 35 students, during normal school hours and in a classroom specially prepared for this purpose. For participants under 18, parents were asked to provide a written informed consent in order for their child to participate in the study. Participants were informed of the confidentiality of their responses and of the voluntary nature of the study. No incentive was provided for their participation. Administration took place under the supervision of the researchers. The study was approved by the research an ethic committee at the University of La Rioja. 

### 2.4. Data Analyses

First, we calculated the internal consistency of the EAV scores. To obtain a measure of the reliability of the scores, we calculated McDonald’s Omega. In addition, we gather evidences of convergence validity by means of the composite reliability (CR) and the average variance extracted (AVE). Values greater than 0.6 and 0.5 are considered adequate for CR and AVE respectively. Moreover, we analyzed the differential validity studying the square root of the AVE value. The data for the diagonal position are the square root of the mean variance extraction rate (AVE value) for each study variable If the square root of the mean variance extraction rate (AVE value) of each question is greater than the correlation coefficient between the variables, it indicates that there is a strong discriminant coefficient between the variables, that is, the difference between each measurement variable is better. Second, in order to analyze the internal structure of the EAV, we performed several confirmatory factor analysis (CFA) in the second subsample. We tested a one-dimensional factor model, a two-factor model, derived from the results of the EFA, and a bifactor solution with a general factor and two group factors. The WLSMV estimator for dichotomous items was used. The following goodness-of-fit indices were used: Chi-square (χ^2^), Comparative Fit Index (CFI), Tucker-Lewis Index (TLI), Root Mean Square Error of Approximation (RMSEA), and Weighted Root Mean Square Residual (WRMR). Hu and Bentler [30] suggested that RMSEA should be. 06 or less for a good model fit and CFI and TLI should be 0.95 or more, though any value over 0.90 tends to be considered acceptable. For WRMR, values less than 0.95 indicate good model fit (for dichotomous outcomes) [31]. Fourth, in order to test measurement invariance across gender, successive multigroup CFAs were conducted. Using Delta parameterization in Mplus, two steps on measuring invariance need to be considered: configural and strong invariance models [32]. The ∆CFI were used to determine in cases where nested models were practically equivalent.

Third, and with the aim to test measurement invariance (MI) by gender and age, successive multigroup CFAs were conducted [33] In order to compare age, we established two different groups: younger adolescents (12–15 years old) and older adolescents (16–19 years old),.Basically, a hierarchical set of steps are followed when MI is tested, typically starting with the determination of a well-fitting multigroup baseline model and continuing with the establishment of successive equivalence constraints in the model parameters across groups. The analyzed dimensional models can be seen as nested models to which constraints are progressively added. Due to the limitations of the ∆χ^2^ regarding its sensitivity to sample size, Cheung and Rensvold [34] proposed a more practical criterion, the change in CFI (∆CFI), to determine if nested models are practically equivalent. In this study, when ∆CFI is greater than 0.01 between two nested models, the more constrained model is rejected since the additional constraints have produced a practically worse fit. However, if the change in CFI is less than or equal to 0.01, it is considered that all specified equal constraints are tenable and, therefore, it is possible to continue with the next step in the analysis of MI. Latent mean differences across gender and age were estimated, fixing the latent mean values to zero in the male and in the younger group respectively. For comparisons among groups in the latent means, statistical significance was based on the *z* statistic. The group in which the latent mean was fixed to zero was considered as the reference group.

Fourth, the associations between self-reported EAV scores and other measures including the EAV and the RADS subscales were examined using Pearson’s correlations. In addition, we conducted a mediation analysis. To this purpose, we followed a two-step procedure [35], adapted to analyses the mediation effect in order to confirm the structural relations of the latent variables. SPSS 24.0 [36], Mplus 7.4 [32], and FACTOR 10.0 [37]) were used for data analysis.

## 3. Results

### 3.1. Descriptive Statistics for All the Measuring Instruments and Evidences of Reliability of the EAV Scores

First, descriptive statistics for the subscales and total scores of the measuring instruments used were calculated (see Table 1). Descriptive statistics of the EAV items are depicted in Table 2. As can be seen in Table 2, the prevalence of attitudes towards violence ranged from 1.52 (item 9 = “when a member of the couple presents excessive consume of substance like alcohol or drugs”) to 1.27 (item 10 = when a member of the couple refuse to have sexual intercourse). No statistically significant differences were found by gender in the EAV total score (*t* = 0.910; *p* = 0.363). Table 3 shows the results of the analysis of the McDonald’s Omega, the CR, the AVE and the root square of the AVE. As it can be seen, CR was above the recommended 0.7 value in all the variables. Also, AVE was higher than 0.50, revealing good evidences of convergent validity. In addition, the correlation between the constructs was lower than the root square of AVE, indicating adequate evidences of discriminant validity. Finally, the McDonald’s Omega was 0.862 and 0.872 for the two hypothesized factors, revealing adequate internal consistency of the scores.

### 3.2. Validity Evidences of Factorial Structure 

The analysis of the EFA in the first subsample revealed statistically significant values of Bartlett’s Sphericity Index (2539.8), being statistically significant (*p* < 0.001). Also, Kaiser–Meyer–Olkin (KMO) indices were above 0.85 in all cases. The GFI values found were in all the dimension above. 95. In addition, the RMSR was under 0.08. A two-factor solution explained more than 35% of the variance in all the dimensions. Factor 1 was composed of items 1, 2, 3, 4, and 5 that are related to justification of the violence due to misbehave of the partner. On the other hand, Factor 2 was integrated by items 6, 7, 8, 9, and 10 which relate to justification of the violence because a history of problems (e.g., emotional problems) of the partner.

After the EFA, we conducted different CFA at the item level. Table 2 shows the goodness-of-fit indices for the different factor models tested. The one-dimensional model yielded adequate CFI and TLI values over. 95, however RMSEA values were over the recommended. 06 cut off value, as well as the WRMR values. Moreover, the bifactor solution revealed poor goodness-of-fit indices. Therefore, we decided to retain the two-factor model as the most adequate solution (see Figure 1). The standardized factor loadings for the whole sample as well as for males and females are shown in Table 2. The range of the factor loadings, for the total sample was from 0.36 (item 1) to 0.72 (item 3). All standardized factor loadings estimated were statistically significant (*p* < 0.01).

### 3.3. Measurement Invariance of the EAV Scores across Gender

Given that the one-factor model evidenced good model fit, we therefore tested the measurement invariance of the EAV scores across gender and age. Prior to the analysis of measurement invariance across gender and age, we tested whether the two-factor model showed a reasonably good fit to the data in each group separately (see Table 4). Goodness-of-fit indices for males and females, as well as for the two age’s groups were adequate. The configural invariance model in which no equality constraints across groups were imposed showed an adequate fit to the data. Next, a strong invariance model was tested with the item thresholds and factor loadings constrained to be equal across groups. The ΔCFI between the constrained and unconstrained models was under 0.01, indicating that strong measurement invariance across gender and age was supported for the EAV scores

### 3.4. Evidences of Relation with Other Variables

We calculated the Pearson’s correlation between EAV total score and the RADS, the M-CTS subscales, and the RADS subscales. As shown in Table 5, statistically significant associations were found between the EAV scores and the M-CTS and the RADS. Specifically, the EAV showed a positive and significant association between the EAV and the M-CTS subscales related to psychological aggression as a victim and all the physical aggression both as a victim and as a perpetrator. In addition, positive significant correlations were found between the EAV and the Anhedonia and Negative subscales of the RADS. 

### 3.5. Mediation Analysis

With the aim to analyze the mediation effect, we used structural equation modeling (SEM). First, the direct effect of the M-CTS scores on attitudes towards violence without mediators was tested. The directly standardized path (β  =  −0.46, *p*  <  0.001) was significant. Then, a partially-mediated model containing a mediator (depression) and a direct path from scores on the M-CTS to attitudes towards violence was tested. All the path coefficients were statistically significant. The results showed an acceptable fit of the model to the data [χ^2^ (*df*  =  15) = 20.35, χ^2^/*df* = 1.19; RMSEA  =  0.036; SRMR  =  0.051 and CFI  =  0.981. These results revealed that scores on the M-CTS and depression have significant effects on attitudes towards violence among adolescents and youths.

Then, the mediating effects of depression on attitudes towards violence and scores of the M-CTS were tested for significance by adopting the Bootstrap estimation procedure in AMOS (a bootstrap sample of 1000 was specified). Table 2 shows the indirect effects and their associated 95% confidence intervals. The indirect effect of the M-CTS on attitudes towards violence through depression was significant.

## 4. Discussion

To date, violence against women and violence manifestations within the context of romantic relationships is becoming a world global issue [3,4,38]. Specially worrying is the fact the IPV is starting earlier, with a larger number of adolescents involved in this kind of violence [5,6,7]. Adolescence intimate partner violence rates are increasing affecting now to more than half of all dating youth [1]. Nonetheless, to date, little is known about attitudes towards intimate partner violence across the world and, specifically, in Spain. Moreover, there is a lack of adequate and sound instruments measuring this.

The present study aimed, thus, to examine the prevalence, factorial structure, measurement invariance across gender, and reliability of the EAV scores, as well as its associations with intimate partner violence and depression symptoms, in a large sample of non-clinical adolescents. The study of evidences of an instrument such as the EAV allows generating and assess profiles of possible adolescents and youth that are more likely to engage in IPV. With this regard, present study reveals that the EAV is a short instrument with adequate evidences of validity and internal consistency of the scores for its use in educational settings like school or university, as well as clinical settings. This is particularly relevant, as it seems reasonable to think that early detection and promotion of positive attitudes towards intimate relationships may prevent IPV.

The results indicated that the EAV is an easy, simple, and brief tool in order to screen for violence attitudes. The study revealed adequate psychometric properties in Spanish adolescents. The internal consistency of the scores estimated by means of ordinal alpha was good. In addition, the results the EAV should be considered as a unidimensional factor structure. Furthermore, this structure was equivalent by gender, after the study of the Measurement Invariance. To date, no previous studies have analyzed, to the best of our knowledge, the factorial structure of the EAV scores in a non-clinical sample of adolescents, being the first study analyzing and the psychometric goodness of the EAV in a Spanish sample. Therefore, future studies should further analyze the extent to which this result are similar in other samples, in order to validate the results found in the present study. A previous study analyzing abuse in non-married couples used the EAV, assuming a unidimensional structure [39], but this assumption had still to be confirmed. Nowadays, there is still, not surprisingly, a lack of studies about the psychometric data on self-report measures, being in their nascent stage [40,41]. Thus, new empirical studies need to replicate the findings established here.

The results revealed that attitudes toward violence were moderate associated with intimate partner violence and with depression symptoms. Specifically, higher rates of attitudes toward violence were associated to psychological aggression as a victim and to both medium and severe physical aggression both as a victim and as a perpetrator. Also, a positive relationship was found between the EAV and Anhedonia and Negative subscales of the RADS. This is somehow consistent with the idea that IPV is related to different issues like trauma and mental health problems, including depression [13,14,15,42]. Worth noting, the correlations found between the RADS and the EAV were low. One possible explanation is that the EAV measures attitudes towards violence instead of IPV *per se.* Future studies should analyze the exact relation between these two constructs. Nonetheless, this study is one of the first posing the association between attitudes towards violence and these mental health issues. In addition, previous studies have established the relation between attitudes toward violence and explicit IPV [43]. With regards to the mediation analysis, the results of the SEM revealed that depression mediated the relationship between attitudes towards violence and scores on the M-CTS. Moreover, the scores on the M-CTS had a statistically significant effect on attitudes towards violence. This is consistent with the idea that those adolescents who justify violence are more likely to engage in IPV, being depression a variable that may affect the outcome. More studies could further analyze this association.

The results of the present study should be interpreted in the light of the following limitations. First, measurement of violence attitudes, as well as depressive symptoms and violence manifestations were based solely on self-report and there are well-known inherent problematics like the effect of stigmatization, the possibility of misunderstanding of some items or the lack of introspection of some participants. Therefore, future studies should consider the use of external informants, interviews or even bio-behavioral and/or biological markers. Second, adolescence is a developmental period in which personality is still consolidating. Thus, the present results must be further evaluated in order to understand their natural developmental course. Third, no information was gathered regarding the participants’ psychiatric morbidity or the use or abuse of substances, aspects that may partially influence the results. Finally, our data was cross-sectional in nature.

Despite the noted limitations, the present study allows confirming the adequate psychometric properties of the EAV, an instrument devoted to assess attitudes towards violence in intimate relationships in a large sample of Spanish adolescents. In addition, these attitudes seem to be related with depressive symptoms and violence manifestations during adolescence. The results have clear implications for the construct validity of the EAV and for its use in school populations in order to study intimate violence attitudes in adolescent populations. In addition, this study contributes relevant information to further understand the structure of and relations of attitudes towards violence, allowing the implementation of future preventive treatments. More research is needed in order to advance in the study of attitudes towards violence in intimate relationships settings and the role that they play in adolescents. Also, the study of measurement invariance of the EAV across other relevant variables like race or culture could also be relevant. In addition, the role of the attitudes toward violence in the prediction and transition to actual violence in intimate relationships during adolescence should continue to be explored in greater depth through independent longitudinal studies.

## Figures and Tables

**Figure 1 ijerph-18-00566-f001:**
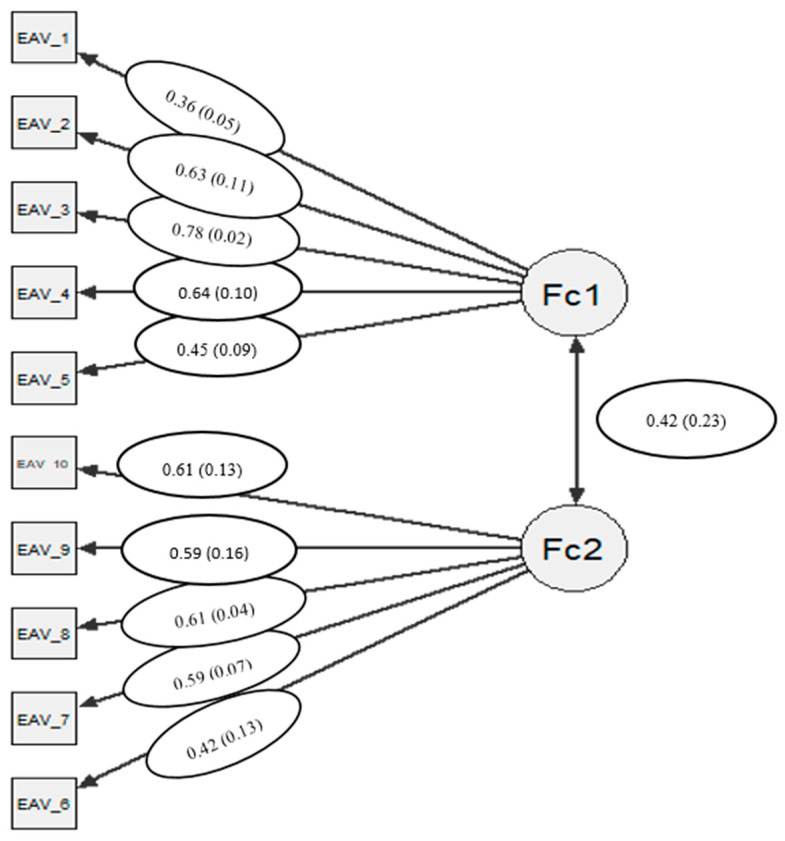
Path diagram of the two factor dimensional model of the Attitudes Scale towards Violence (EAV). Note. In circles factor loadings (measurement errors).

**Table 1 ijerph-18-00566-t001:** Descriptive statistics for the Attitudes Scale Towards Violence (EAV), the Modified Conflict Tactics Scale (MCTS), and the Reynolds (RADS) Subscales for the total sample and across gender.

Scales and Subscales	Total Sample		Males		Females	
	Mean	SD	Mean	SD	Mean	SD
EAV_TOTAL	13.99	6.94	14.22	6.95	13.85	6.94
M-CTS						
Argumentation Victim	8.25	2.71	8.16	2.79	8.30	2.66
Argumentation Perpetrator	7.87	2.55	7.83	2.69	7.89	2.47
Psychological Aggression Victim	11.81	4.17	10.29	3.46	12.65	4.29
Psychological Aggression Perpetrator	11.02	3.78	10.48	3.73	11.32	3.77
Medium Physical Aggression Victim	8.53	3.25	8.26	3.06	8.68	3.35
Medium Physical Aggression Perpetrator	8.31	2.98	8.49	3.39	8.21	2.72
Severe Physical Aggression Victim	3.14	0.99	3.25	1.44	3.09	0.66
Severe Physical Aggression Perpetrator	3.11	0.77	3.11	0.83	3.11	0.74
Reynolds						
RADS Dysphorya	15.87	4.38	14.67	4.01	16.63	4.43
RADS Anhedonia	11.46	2.85	11.41	2.94	11.49	2.80
RADS Negative	12.63	4.20	12.09	3.77	12.96	4.42
RADS Somatization	15.38	3.44	14.63	3.33	15.85	3.43

**Table 2 ijerph-18-00566-t002:** Descriptive statistics and standardized factor loadings for the Attitudes Scale towards Violence (EAV) items.

			Standardized Factor Loadings				
EAV Items	M	SD	Total Sample	Male	Female	13–16 Years Old	18–21 Years Old
Factor 1							
In what circumstances do you consider the use of intimate partner violence justified?							
When a member of the couple is unfaithful	1.40	0.92	0.36 (0.20)	0.47 (0.22)	0.46 (0.21)	0.45 (0.22)	0.39 (0.14)
When a member of the couple disqualifies the other in front of third parties	1.40	0.91	0.63 (0.36)	0.63 (0.40)	0.62 (0.38)	0.64 (0.45)	0.51 (0.37)
When one member of the couple disqualifies the other in front of his/her family	1.42	0.96	0.78 (0.61)	0.83 (0.69)	0.75 (0.56)	0.71 (0.60)	0.69 (0.62)
When a member of the couple insults the other	1.42	0.97	0.64 (0.41)	0.63 (0.40)	0.66 (0.43)	0.53 (0.32)	0.75 (0.43)
The use of violence is not justified under any circumstances	1.34	0.93	0.45 (0.24)	0.43 (0.31)	0.56 (0.32)	0.40 (0.18)	0.38 (0.23)
Factor 2							
In couples with little education	1.41	0.97	0.42 (0.38)				
When one or both members of the couple has a personal history of abuse or has witnessed violence in the family of origin	1.40	0.92	0.59 (0.35)	0.61 (0.37)	0.59 (0.36)	0.60 (0.47)	0.48 (0.34)
When one or both members of the couple present emotional alterations such as impulsivity, anxiety, depression	1.41	0.87	0.61 (0.39)	0.49 (0.35)	0.66 (0.44)	0.60 (0.36)	0.63 (0.38)
When one or both members of the couple have abusive use of substances such as alcohol and/or drugs.	1.52	1.05	0.59 (0.34)	0.58 (0.34)	0.59 (0.35)	0.59 (0.35)	0.56 (0.35)
When one of the members of the couple refuses to have sex.	1.27	0.81	0.61 (0.38)	0.57 (0.33)	0.65 (0.42)	0.60 (0.37)	0.60 (0.36)

Note. M = Mean; SD = Standard Deviation.

**Table 3 ijerph-18-00566-t003:** Evidences of composite reliability, average variance extracted for the Attitudes Scale Towards Violence (EAV) items.

Factors	AVE	CR	McDonald’s Omega	1	2
1	0.729	0.896	0.862	0.854 *	
2	0.766	0.903	0.872	0.721 **	0.875 *

Note. AVE = average variance extract. * = square root of the variance shared between the constructs and their measures. ** = correlations among the constructs. For evidences of discriminant validity, the correlations among constructs should be lower than the square root of the variance shared.

**Table 4 ijerph-18-00566-t004:** Goodness-of-fit indices for the hypothetical models tested and measurement invariance across gender and age.

Model	χ^2^	*df*	CFI	TLI	RMSEA (90% IC)	WRMR	ΔCFI
One-factor	512.869	68	0.984	0.982	0.073 (0.068–0.079)	2.267	
Two factor model	268.564	64	0.986	0.991	0.042 (0.038–0.046)	0.756	
Bifactor model	850.469	45	0.901	0.895	0.091(0.085–0.096)	2.954	
Measurement Invariance (Two factor model)							
Male (*n* = 483)	239.182	35	0.984	0.990	0.043 (0.040–0.049)	0.332	
Female (*n* = 765)	387.739	35	0.985	0.992	0.040 (0.038–0.043)	0.568	
Configural invariance	332.615	70	0.983	0.975	0.039 (0.035–0.043)	0.225	
Strong invariance	317.549	108	0.989	0.991	0.041 (0.038–0.044)	0.333	−0.01
13–16 years old	180.245	35	0.985	0.984	0.043 (0.039–0.048	0.410	
17–21 years old	198.065	35	0.983	0.985	0.042 (0.036–0.046)	0.405	
Configural invariance	240.689	70	0.988	0.989	0.041 (0.037–0.044)	0.352	
Strong invariance	310.436	108	0.988	0.990	0.040 (0.035–0.043)	0.398	−0.01

Note. χ^2^ = Chi square; *df* = degrees of freedom; CFI = Comparative Fit Index; TLI = Tucker-Lewis Index; RMSEA = Root Mean Square Error of Approximation; IC = Interval Confidence; WRMR = Weighted Root Mean Square Residual; ΔCFI = Change in Comparative Fix Index.

**Table 5 ijerph-18-00566-t005:** Pearson’s Correlations between the Attitudes Scale Towards Violence (EAV), the Modified Conflict Tactics Scale (MCTS), and the Reynolds (RADS) Subscales.

	1	2	3	4	5	6	7	8	9	10	11	12	13
EAV_TOTAL (1)	-												
M-CTS Arg A (2)	0.05	-											
M-CTS Arg B (3)	0.01	0.73 **	-										
M-CTS Psy Aggre A (4)	0.09 **	0.13 **	0.19 **	-									
M-CTS Psy Aggre B (5)	0.05	0.19 **	0.23 **	0.76 **	-								
M-CTS Med Phy Aggre A (6)	0.25 **	−0.03	0.01	0.35 **	0.33 **	-							
M-CTS Med Phy Aggre B (7)	0.17 **	0.01	0.03	0.26 **	0.36 **	0.75 **	-						
M-CTS Sev Phy Aggre A (8)	0.35 **	−0.08	-0.05	0.10 *	0.13 *	0.57 **	0.49 **	-					
M-CTS Sev Phy Aggre B (9)	0.26 **	−0.08	-0.08	0.09	0.10	0.36 **	0.43 **	0.71 **	-				
RADS Dysphorya (10)	0.01	0.09 **	0.10 **	0.32 **	0.26 **	0.09 **	0.13 **	−0.04	0.02	-			
RADS Anhedonia (11)	0.07 **	−0.03	0.04	0.08 *	0.07 *	0.07	0.07 *	0,02	0.02	0.39 **	-		
RADS Negative (12)	0.06 *	0.07 *	0.08 *	0.30 **	0.23 **	0.15 **	0.18 **	0.06	0.04	0.70 **	0.44 **	-	
RADS Somatization (13)	0.01	0.07*	0.08*	0.30 **	0.28 **	0.14 **	0.13 **	0.10	0.08	0.60 **	0.25 **	0.50 **	-

Note. M-CTS Arg A Argumentation as Victim; M-CTS Arg B = Argumentation as Perpetrator; M-CTS Psy Aggre A = Psychological Aggression as Victim; M-CTS Psy Aggre A = Psychological Aggression as Perpetrator; M-CTS Med Phy Aggre A = Medium Physical Aggression as Victim; M-CTS Med Phy Aggre B = Medium Physical Aggression as Perpetrator; MCTS Sev Phy Aggre A: Severe Physical Aggression as Victim; MCTS Sev Phy Aggre B: Severe Physical Aggression as Perpetrator. ** *p* < 0.01; * *p* < 0.05.

## Data Availability

The data is not available as it contains sensible information about human people and consent was not obtained for its dissemination.

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
