# Peer review of "Attitudes towards Violence in Adolescents and Youth Intimate Partner Relationships: Validation of the Spanish Version of the EAV"

_ijerph, 2021, doi:10.3390/ijerph18020566_

Round 1

Reviewer 1 Report

It is a good job, very precise, adequate, interesting or current. However, some improvements are suggested:

1) Include more current sources (2020), both in the background and in the discussion

2) Include data for composite reliability (eg, Omega McDonald), discriminant validity, convergent validity (mean variance extracted)

3) it would be convenient to provide a brief description of the results in the exploratory factor analysis, and to what extent a single factor was obtained; Other options should be tested in the AFC. Is a model with only one factor better than with two, for example?

4) The instrument measurement model diagram should be included

5) The statements of the items in table 2 should be included instead of the item number

6) Composite reliability data, mean variance extracted, convergent and discriminant validity of the other two instruments used, with the data of the present study, must be provided.

7) In the background the research question must be clearly stated, which is specified in the objective and which is materialized with the hypotheses of the study

8) The structural validity (interrelation of the current scale with the others) should be assessed and perhaps a SEM structure model should be proposed with them and their degree of adjustment should be assessed.

9) A paragraph should be included in discussions about the implications and practical applications of the study

It is a good job, however the suggestions indicated must be followed to give it the rigor and added value intended. Recommendation: moderate changes

Author Response

We have provided a point-by-point answer to all the suggestions raised. 

We would like to thank the reviewer for all the relevant comments and suggestions provided. We have taken all of them into consideration and modified consequently the manuscript. Here it is possible to see all the points raised and the answers provided. We truly believe that the manuscript has been clearly improved.

Reviewer 1.

1) Include more current sources (2020), both in the background and in the discussion

Reply: According to the reviewer suggestion, we have introduced the following paragraph in the new version of the manuscript:

Serrano-Montilla, C.; Lozano, L.M.; Bender, M.; Padilla, J.L. Individual and societal risk factors of attitudes justifying intimate partner violence against women: A multilevel cross-sectional study. BMJ Open 2020, 10.

Boyce, S.C.; Deardorff, J.; Minnis, A.M. Relationship Factors Associated With Early Adolescent Dating Violence Victimization and Perpetration Among Latinx Youth in an Agricultural Community. J. Interpers. Violence 2020.

Sanz-Barbero, B.; Pereira, P.L.; Barrio, G.; Vives-Cases, C. Intimate partner violence against young women: Prevalence and associated factors in Europe. J. Epidemiol. Community Health 2018, 72, 611–616.

Serrano-Montilla, C.; Valor-Segura, I.; Padilla, J.L.; Lozano, L.M. Public helping reactions to intimate partner violence against women in European countries: The role of gender-related individual and macrosocial factors. Int. J. Environ. Res. Public Health 2020, 17, 1–18.

2) Include data for composite reliability (eg, Omega McDonald), discriminant validity, convergent validity (mean variance extracted)

Reply: We think that this comment is interesting. Although the paper had analyzed different sources of validity with other variables, as well as measurement invariance, and evidences of internal structure, we have analyzed the composite reliability, as well as mean variance extracted and discriminant validity (see new table 3).

The following has been added in the manuscript:

“Table 3 shows the results of the analysis of the McDonald’s Omega, the CR, the AVE and the root square of the AVE. As it can be seen, CR was above the recommended 0.7 value in all the variables. Also, AVE was higher than 0.50, revealing good evidences of convergent validity. In addition, the correlation between the constructs was lower than the root square of AVE, indicating adequate evidences of discriminant validity. Finally, the McDonald’s Omega was 0.862 and 0.872 for the two hypothesized factors, revealing adequate internal consistency of the scores.”

3) it would be convenient to provide a brief description of the results in the exploratory factor analysis, and to what extent a single factor was obtained; Other options should be tested in the AFC. Is a model with only one factor better than with two, for example?

Reply: We think that the reviewer comment about this point is relevant. The present study did not explore the EFA approach. Following the reviewer comment, we have made a cross validation study and we have performed EFA in the first subsample and CFA in the second subsample. In addition, we have performed more CFA than the one studied in the first version.

4) The instrument measurement model diagram should be included

Reply: Attending to the reviewer suggestion, the model diagram has been included in the new version of the manuscript. (Please see new Figure 1):

5) The statements of the items in table 2 should be included instead of the item number

Reply: Attending to the reviewer comment, the statement of the items have been included in table 2.

6) Composite reliability data, mean variance extracted, convergent and discriminant validity of the other two instruments used, with the data of the present study, must be provided.

Reply: Attending to the reviewer comment, we have included composite reliability and mean variance extracted for the other two instruments, which are evidences of convergent validity.

7) In the background the research question must be clearly stated, which is specified in the objective and which is materialized with the hypotheses of the study

Reply: We think that the reviewer comment is relevant. We have added the following in the introduction to justify the hypothesis:

“The EAV encompass ten items addressing the degree to which the individual considers appropriate the use of violence towards the partner in different situations. Validity evidences of the EAV were calculated, as well as the reliability of the scores. Nonetheless, the psychometric properties of this recent version have not been, yet, reported. Therefore, a question needs still to be solve. Is it possible to use the EAV in its Spanish version as an instrument with adequate evidences of reliability of the scores and validity?  “

8) The structural validity (interrelation of the current scale with the others) should be assessed and perhaps a SEM structure model should be proposed with them and their degree of adjustment should be assessed.

Reply: We think that the comment is interesting. In the present version of the manuscript, evidences of validity with other variables have been assessed. Also, and considering previous comments of the reviewer, we have conducted a EFA, in addition of new CFA, as noted in comment. While not being equivalent to SEM, we think that with this new approach, the use of SEM could not be that relevant. In any case, we are open to discuss this point

9) A paragraph should be included in discussions about the implications and practical applications of the study

Reply: Attending to the reviewer suggestion, we have included a paragraph about implications and applications of the present study:

“The study of evidences of an instrument such as the EAV allows generating and assess profiles of possible adolescents and youth that are more likely to engage in IPV. With this regard, present study reveals that the EAV is a short instrument with adequate evidences of validity and internal consistency of the scores for its use in educational settings like school or university, as well as clinical settings. This is particularly relevant, as it seems reasonable to think that early detection and promotion of positive attitudes towards intimate relationships may prevent IPV.”

Reviewer 2 Report

The research topic is extremely important and meets the requirement of novelty and added value. It was interesting to read the manuscript, and I liked that the ideas are presented in a structured and neat manner. Some minor suggestions:

  • I would suggest renaming “Reynolds” (Reynolds subscales, everywhere in the tables and text) to “RADS”.
  • I would suggest separate analyses for 13 - <18 years old, and 18> - 21 years old respondents.
  • Please, explain, why you have chosen these specific instruments for validation of EAV;
  • EAV and RADS demonstrated extremely low correlations. How would you explain that?
  • What prevented you from surveying possible associations between, for example, EAV and personal experience?
  • I would suggest changing the title of the manuscript “Attitudes towards violence in adolescence []”, as you analyzed data of the 18-21 years old respondents who do not belong to the adolescents’ group.
  • I would suggest adding more information on the test (EAV) itself, as it might have an added value for other researchers.
  • I would suggest clearly presenting the results of CFA.

Anyway, I appreciate the great work that the researchers have done, and I recommend to publish the manuscript after minor revisions.

Author Response

We would like to thank the reviewer for all the relevant comments and suggestions provided. We have taken all of them into consideration and modified consequently the manuscript. Here it is possible to see all the points raised and the answers provided.

Reviewer 2

The research topic is extremely important and meets the requirement of novelty and added value. It was interesting to read the manuscript, and I liked that the ideas are presented in a structured and neat manner. Some minor suggestions:

  • I would suggest renaming “Reynolds” (Reynolds subscales, everywhere in the tables and text) to “RADS”.

We appreciate the reviewer suggestion. We have modified Reynolds for RADS as suggested.

  • I would suggest separate analyses for 13 - <18 years old, and 18> - 21 years old respondents.

We think that the reviewer comment is interesting. We have conducted new analyses (eg. MI), analyzing the age. The suggestion is relevant, nonetheless, we have considered that 13-16 and 17-21 attending to Salmera-Aro (2011) are two very differentiate stages of adolescence where executive functions develop, so we have decided to used these two age’s ranges. In any case, we are open to change it if necessary.

  • Please, explain, why you have chosen these specific instruments for validation of EAV;

EAV and RADS demonstrated extremely low correlations. How would you explain that?

Attending to the suggestion, we have added some new information about the evidences of relationship between gender violence and emotional problems, which led us to use the RADS for this purpose. One possible explanation is that the questionnaire measures attitudes towards violence instead of violence per se. This, has been also added in the discussion section. The following has been added in the introduction and in the discussion sections:

Introduction:

“With this regard, new attention is being devoted to the quality of romantic relationships and the ideas and attitudes of adolescents to this in order to promote healthy relationships [25], and the fact that these ideas could be related to emotional symptoms, as well as intimate partner relationships.”

Discussion:

“Worth noting, the correlations found between the RADS and the EAV were low. One possible explanation is that the EAV measures attitudes towards violence instead of IPV per se. Future studies should analyze the exact relation between these two constructs”

  • What prevented you from surveying possible associations between, for example, EAV and personal experience?

We think that the reviewer comment is relevant. We decided to analyze certain variables as the one explained in the document and some others not included in the present article. As the reviewer suggest, it would have been relevant to include some others variables but this would have implied more variables in the study, that was long enough. Nonetheless, and attending to the comment, this has been included in the limitations section.

  • I would suggest changing the title of the manuscript “Attitudes towards violence in adolescence []”, as you analyzed data of the 18-21 years old respondents who do not belong to the adolescents’ group.

This is an interesting comment. Although the consideration for adolescence stage vary across authors, being till the age of 26 in some sound investigations, we have changed the tittle to attitudes towards violence in adolescents and youth to make it more inclusive

  • I would suggest adding more information on the test (EAV) itself, as it might have an added value for other researchers.

Considering the reviewer comment, we have new information about the test in the introduction and the items of the questionnaire have been included in Table 2 (also suggested by reviewer 1).

The following as been added in the introduction:

“There is, thus, a general question asking: In what circumstances do you consider the use of intimate partner violence justifie? And then ten different options such as: when a member of the couple is unfaithful (ítem 1) or when a member of the couple disqualifies the other in front of his/her family (ítem 3).”

  • I would suggest clearly presenting the results of CFA.

We think that this comment is appropriate. We have performed new CFA analyses and new results are presented in Table 4.

The following has been added in the results section:
“The analysis of the EFA in the first subsample revealed statistically significant values of Bartlett’s Sphericity Index (2539.8), being statistically significant (p < .001). Also, Kaiser–Meyer–Olkin (KMO) indices were above 0.85 in all cases. The GFI values found were in all the dimension above .95. In addition, the RMSR was under .08. A two factor solution explained more than 35% of the variance in all the dimensions. Factor 1 was composed of items 1, 2, 3, 4, and 5 that are related to justification of the violence due to misbehave of the partner. On the other hand Factor 2 was integrated by items 6,7,8,9, and 10 which relate to justification of the violence because a history of problems (e.g. emotional problems) of the partner.

After the EFA, we conducted different CFA at the item level. Table 2 shows the goodness-of-fit indices for the different factor models tested. The one-dimensional model yielded adequate CFI and TLI values over .95, however RMSEA values were over the recommended .06 cut off value, as well as the WRMR values. Moreover, the bifactor solution revealed poor goodness-of-fit indices. Therefore, we decided to retain the two factor model as the most adequate solution”

Anyway, I appreciate the great work that the researchers have done, and I recommend to publish the manuscript after minor revisions.

We would like to thank the reviewer for these words. We think that the document has been improved with the suggestions made.

Round 2

Reviewer 1 Report

In general, the authors have made the changes indicated in the initial review, so it seems appropriate to consider it ready for publication.

1) It is simply suggested that the entire manuscript be revised and errata be corrected;

2) and the figure of the diagram of the two factors includes the coefficients in each arrow that contribute to the weight of each factor, as well as the measurement errors of each indicator in relation to each latent variable.

3) In. addition, it would be convenient to include a mediation / moderation analysis between the indicated measures, as suggested in the first review, either in the form of SEM or a mediation-moderation analysis between the measures used, in order to see the direct and indirect predictive effect of each of them on the others and determine the best structural pattern and therefore that provides added value beyond the validation of the instrument included

These changes must be marked in color in the text of the article for verification.
These changes can be made in the editing process for publication.

Author Response

We would like to thank the reviewer for the new suggestions.

In general, the authors have made the changes indicated in the initial review, so it seems appropriate to consider it ready for publication.

  • It is simply suggested that the entire manuscript be revised and errata be corrected;

The paper has been revised and different tipos have been corrected

  • The figure of the diagram of the two factors includes the coefficients in each arrow that contribute to the weight of each factor, as well as the measurement errors of each indicator in relation to each latent variable.

The information about the coefficients and the measurement errors has been included.

  • addition, it would be convenient to include a mediation / moderation analysis between the indicated measures, as suggested in the first review, either in the form of SEM or a mediation-moderation analysis between the measures used, in order to see the direct and indirect predictive effect of each of them on the others and determine the best structural pattern and therefore that provides added value beyond the validation of the instrument included

As suggested, a mediation analysis between the measures used has been included. The following has been added in the text:

Data analysis:

In addition, we conducted a mediation analysis. To this purpose, we followed a two-step procedure [33], adapted to analyses the mediation effect in order to confirm the structural relations of the latent variables.

Results:

Mediation analysis

With the aim to analyze the mediation effect, we used structural equation modeling (SEM). First, the direct effect of the M-CTS scores on attitudes towards violence without mediators was tested. The directly standardized path (β = -0.46, p < 0.001) was significant. Then, a partially-mediated model containing a mediator (depression) and a direct path from scores on the M-CTS to attitudes towards violence was tested. All the path coefficients were statistically significant. The results showed an acceptable fit of the model to the data [χ2 (df = 15) = 20.35, χ2/df = 1.19; RMSEA = 0.036; SRMR = 0.051 and CFI = 0.981]. These results revealed that scores on the M-CTS and depression have significant effects on attitudes towards violence among adolescents and youths.

Then, the mediating effects of depression on attitudes towars violence and scores of the M-CTS were tested for significance by adopting the Bootstrap estimation procedure in AMOS (a bootstrap sample of 1,000 was specified). Table 2 shows the indirect effects and their associated 95% confidence intervals. The indirect effect of the M-CTS on attitudes towards violence through depression was significant.

Discussion

With regard to the mediation analysis, the results of the SEM revealed that depression mediated the relationship between attitudes towards violence and scores on the M-CTS. Moreover, the scores on the M-CTS had a statistically significant effect on attitudes towards violence. This is consistent with the idea that those adolescents who justify violence are more likely to engage in IPV, being depression a variable that may affect the outcome. More studies could further analyze this association.